# Verbenalin Reduces Amyloid-Beta Peptide Generation in Cellular and Animal Models of Alzheimer’s Disease

**DOI:** 10.3390/molecules27248678

**Published:** 2022-12-08

**Authors:** Juhee Lim, Seokhee Kim, Changhyun Lee, Jeongwoo Park, Gabsik Yang, Taehan Yook

**Affiliations:** 1College of Pharmacy and Research Institute of Pharmaceutical Sciences, Woosuk University, Wanju 55338, Republic of Korea; 2College of Korean Medicine, Woosuk University, Jeonju 54986, Republic of Korea; 3College of Pharmacy and Research Institute of Pharmaceutical Sciences, Seoul National University, Seoul 08826, Republic of Korea

**Keywords:** Alzheimer’s disease (AD), amyloid beta, brain-derived neurotrophic factor, tau, verbenalin

## Abstract

Verbenalin, among the major constituents of *Verbena officinalis*, has been reported to exhibit sleep-promoting and antioxidant activities. This study demonstrates the effects of verbenalin on amyloid-beta (Aβ) peptide generation in Swedish mutant amyloid precursor protein (APP)-overexpressing Neuro2a cells (SweAPP/N2a) and in Alzheimer’s disease (AD) animal models. We further performed molecular biological analyses of these in vitro and in vivo models of AD. The effects of verbenalin were assessed based on the expression of factors related to Aβ peptide production using Western blotting, enzyme-linked immunosorbent assay, and immunohistochemistry (IHC). The intracellular expression and release of APP protein were both decreased by verbenalin treatment in SweAPP/N2a cells. Thus, the production of Aβ peptides was decreased. Compared to those in AD transgenic (Tg) mice, IHC revealed that verbenalin-treated animals showed decreased Aβ and tau expression levels in the hippocampus. In addition, verbenalin restored the expression of brain-derived neurotrophic factor (BDNF) in the hippocampus of AD animal models. These findings suggest that verbenalin may decrease Aβ formation both in vitro and in vivo. Verbenalin may also help improve the pathological hallmarks of AD.

## 1. Introduction

Alzheimer’s disease (AD) is the most common chronic neurodegenerative disease and the leading cause of dementia worldwide. It is characterized by a reduction in thinking ability and independence in daily activities [1,2]. Hippocampal atrophy and ventricle enlargement are anatomical markers of AD development as the hippocampus plays a major role in cognitive function, learning, and memory [3]. Despite the growing prevalence of AD, few FDA-approved medications are available on the market; further, these do not reverse or halt the degeneration or death of neurons that lead to AD. The typical neuropathological alterations in AD include the formation of intracellular neurofibrillary tangles and extracellular amyloid plaques [4]. The plaques are composed of amyloid beta (Aβ), which is generated from amyloid precursor protein (APP), while neurofibrillary tangles (NFTs) are composed of hyperphosphorylated tau protein. After being produced as a soluble monomer, Aβ forms various intermediate aggregation states, such as dimers and trimers, soluble aggregates, and protofibrils. Of these, the soluble aggregates are associated with dementia [5]; it is thus widely accepted that Aβ is essential for the initiation and development of AD [6]. Given the pathogenesis of AD, numerous studies are being conducted to develop therapeutic agents to reduce the accumulation of Aβ [7,8]. Aside from the traditional Aβ and tau hypotheses, cerebral hypoperfusion caused by vascular dysfunction has recently been identified as among the causative factors for AD onset [9]. Aβ deposition is known to increases with age and progressive Aβ deposition in the brain is associated with arterial stiffness [10]. Furthermore, ischemic damage caused by abnormal blood flow to the brain can disrupt or reduce oxygen supply to brain tissues, resulting in extensive nerve cell destruction [11]. Therefore, considering the multifactorial nature of AD with complex pathophysiological causes, treatment strategies must be developed using a multifaceted approach.

Verbenalin, a natural product found in *Verbena officinalis*, is classified as an iridoid glucoside and has a wide range of biological activities, including sleep-inducing, antioxidant, and hepatoprotective properties [12,13,14]. In particular, aqueous extracts from *Verbena officinalis* have been reported to decrease Aβ-induced neurotoxicity such as neurite shorting and DNA condensation in primary cortical neurons [15]. In a recent study, human neuroblastoma SH-SY5Y cells treated with verbenalin showed neuroprotective properties against Aβ-induced cytotoxicity [16]. Furthermore, a microarray analysis of verbenalin-treated human amnion epithelial cells revealed changes in genes associated with AD, indicating its therapeutic potential [16].

This study investigated the effects of verbenalin on Aβ peptide production in vitro and in vivo. The effect of verbenalin the expression and secretion of APP proteins and Aβ peptide generation was investigated. In addition, verbenalin has previously restored the expression of brain-derived neurotrophic factor (BDNF) in the hippocampal region of AD animal models. Recent studies have shown that serum BDNF can be used as a biomarker for AD diagnosis in clinical practice since it reflects Aβ aggregation in the brain [17,18]. However, additional research is still necessary to determine the cognitive enhancing effects of verbenalin, as well as its precise mechanism of action in animal models.

## 2. Results

The pathological hallmark of AD is Aβ deposition in the brain; endoproteolytic processing of APP results in either neurotoxic Aβ or secreted ectodomain APPα. First, we examined APP protein expression levels in SweAPP/N2a cells. After 24 h of treating SweAPP/N2a cells with verbenalin, immunoblotting data revealed that APP expression was downregulated in the cell lysate and supernatant (Figure 1). Further, both forms of Aβ were evaluated using ELISA, and verbenalin was found to significantly decrease Aβ 42 levels, though Aβ 40 levels showed no significant change (Figure 2).

Next, we investigated how verbenalin decreases Aβ peptide generation in an AD animal model. In particular, the hippocampus is among the first brain regions to be affected by AD, and hippocampal dysfunction is thought to underpin the key features of cognitive impairment. However, measuring hippocampal volume alone is not always sufficient to detect early AD [19]. The cornu ammonis areas (CA1–4), dentate gyrus (DG), and subiculum are interconnected subregions of the hippocampus with distinct histological characteristics and special functions [20].

As shown in Figure 3, the effects of verbenalin on specific AD-related hallmarks were evaluated using IHC. Aβ-positive CA1 and CA2 fibers in C57BL/6J mice displayed a modest reaction (+), resulting in an overall immunostaining intensity of 2. Compared to those in C57BL/6J mice, the nerve fibers of B6Cg-Tg mice showed a strong Aβ-positive reaction (+++) in CA1, polymorphic layer of the DG (PoDG), and DG; a moderate response (++) in CA2; a weak response (+) in CA1; and an overall immunostaining intensity of 12. Donepezil treatment resulted in a strong Aβ-positive (+++) reaction only in the DG, and a weak (+) or no response in the remaining CA1, CA2, CA3, and PoDG. The overall immunostaining intensity was 5. Except for CA2, the group administered verbenalin at 100 mg/kg demonstrated strong immunostaining similar to that in the B6Cg-Tg mice in all subregions (CA1, PoDG, and DG), resulting in an overall immunostaining intensity of 10. Comparison to B6Cg-Tg mice, the mice administered verbenalin at 200 mg/kg showed a weak (+) or no reaction (-) in all regions, resulting in an overall immunostaining intensity of 1 (Table 1).

Another pathological hallmark of AD is tau expression. Tau is a phosphoprotein abundant in mature neuronal axons and regulates microtubule stability. Hyperphosphorylated tau protein in NFTs can disrupt the microtubule network, impairing neuronal function and viability [21]. IHC staining of tau in the hippocampus is shown in Figure 3B. The overall immunostaining intensity in C57BL/6J mice was 0, as no immunostaining response was observed in any region. In B6Cg-Tg mice, compared with C57BL/6J mice, CA3 showed a strong immunostaining response (+++), CA1, CA2, and PoDG showed a moderate response (++), while DG showed a weak response (+). The overall intensity of immunostaining was 10. The donepezil treatment group showed a strong immunostaining response (+++) only in CA3 cells and a weak response (+) in PoDG cells, CA1, and CA2, while DG did not exhibit any immunostaining; therefore, the overall immunostaining intensity was 4. In CA3 cells, verbenalin produced strong (+++) and moderate (++) responses. However, at other sites (CA1, CA2, PoDG, and DG), there was either a weak response (+) or no response (-), resulting in overall immunostaining intensities of 4 (verbenalin 100 mg/kg) and 2 (verbenalin 200 mg/kg) (Table 2).

Next, we attempted to determine how verbenalin reduced Aβ and tau expression in the hippocampus of B6Cg-Tg mice. BDNF is a neurotrophin, a protein that promotes the survival, function, and development of neurons [22,23]. Although BDNF deficiency has been linked with Aβ accumulation and tau phosphorylation, the precise mechanisms underlying the effects of impaired BDNF signaling on AD remain unknown. BDNF-positive cells in the hippocampus were stained using IHC (Figure 4).

In C57BL/6J mice, the PoDG and DG showed a moderate (++) staining response; therefore, the overall immunostaining intensity was 4. B6Cg-Tg mice showed a weak (+) response in DG and PoDG, resulting in an overall immunostaining intensity of 2. Donepezil-treated B6Cg-Tg mice demonstrated a more robust moderate (++) immunostaining response in the PoDG and DG, with an overall immunostaining intensity of 4. Compared to that in B6Cg-Tg mice, verbenalin administered animals displayed weak (+) immunostaining responses in CA1 and CA3 and moderate (++) immune responses in PoDG and DG. The overall immunostaining intensity was 6 (Table 3).

## 3. Discussion

Aβ peptides are produced via APP proteolysis by β- and γ-secretases, and newly generated Aβ is either released into the extracellular environment or remains associated with the plasma membrane [24]. Accumulation of Aβ peptides in the brain has been proposed as an early toxic event in the development of AD. There are two major isoforms of Aβ: the 42-residue Aβ 42 and 40-residue Aβ 40. The most abundant Aβ isoform in the brain is Aβ 40, whereas Aβ 42 shows a significant increase in certain forms of AD [25]. Numerous studies have demonstrated that Aβ 42 has a variety of toxic mechanisms both in vivo and in vitro, including excitotoxicity, mitochondrial changes, synaptic dysfunction, abnormal calcium homeostasis, and oxidative stress [24,25].

In the present study, verbenalin was chosen among the components of various herbal extracts for its ability to inhibit Aβ production (data not shown). Verbenalin, an iridoid glucoside, was shown to reduce both the intracellular and extracellular expression of APP, which is a precursor to Aβ (Figure 1). Furthermore, as shown in Figure 2A, verbenalin markedly reduced Aβ 42 in SweAPP/N2a cells. Numerous investigations have demonstrated that the hippocampus and amygdala, as well as other structures of the medial temporal lobe show significant atrophy as AD progresses [26,27,28]. As each subregion of the hippocampus has different neuropathological vulnerabilities and functions, we separately examined the immunoreactivity of Aβ, tau, and BDNF in hippocampal subregions [19,29]. Although we attempted to investigate the immunoreactivity of Aβ, tau, and BDNF in hippocampal subregions, the subregions showing a closer association with AD pathology remain unclear [19]. In this study, the IHC immunostaining intensity of each area was determined based on the overall immunostaining intensity. As shown in Figure 3 and Table 1 and Table 2, verbenalin reduced the expression of Aβ and tau in the hippocampus of AD animal models, which was consistent with the results of the in vitro assays. Taken together, these findings suggest that verbenalin can be used as an adjuvant therapy for AD owing to its ability to improve AD pathology associated with Aβ and tau. However, verbenalin cannot yet be regarded as a therapeutic tool because there have been no studies on its pharmacodynamic selectivity and no findings about its safety.

## 4. Materials and Methods

### 4.1. Cell Cultures

Swedish mutant APP overexpressing Neuro2a (SweAPP/N2a) cells were provided by Prof. Jae Yoon Leem (Woosuk University). Cells were cultured in Dulbecco’s modified Eagle medium (DMEM) and Opti-MEM supplemented with 5% fetal bovine serum, 1% penicillin/streptomycin amphotericin B, and 1% L-glutamine in 5% CO_2_ at 37 °C. Verbenalin was purchased from Sigma–Aldrich (St. Louis, MO, USA). SweAPP/N2a cells were plated in wells of plates of variable size and treated with a range of verbenalin concentrations.

### 4.2. Western Blotting

SweAPP/N2a cells were washed with PBS and incubated with lysis buffer containing 10 mM Tris-HCl (pH 7.5), 100 mM NaCl, 1 mM EDTA, 10% glycerol, 1% Triton X-100, 30 mM sodium pyrophosphate, 5 mM glycerol-2-phosphate, 1 mM sodium orthovanadate, 1 mM sodium fluoride, protease inhibitor, and phosphatase inhibitors (Millipore, Burlington, MA, USA) for 30 min on ice. The cellular lysates were centrifuged at 20,000× *g* for 15 min at 4 °C to remove debris. Proteins were then separated using sodium dodecyl sulfate-polyacrylamide gel electrophoresis and transferred to nitrocellulose membranes. Subsequently, the membranes were incubated with 5% skim milk for 1 h and then with an APP antibody (Abcam, Cambridge, UK), followed by HRP-conjugated secondary antibodies (Cell Signaling Technology, Beverly, MA, USA). Specific proteins were visualized using enhanced chemiluminescent HRP substrate (Millipore) and analyzed using a luminescent image analyzer, LAS3000-mini (Fujifilm, Tokyo, Japan). The relative intensity of each band was measured using the Multi-Gauge V 3.0 (Fujifilm). β-Actin was used as the loading control.

### 4.3. Supernatant Immunoblot Analysis

After collecting the culture media, the supernatants were centrifuged at 2000 rpm for 5 min to remove residual cells. The supernatant was then transferred to a new microcentrifuge tube and mixed with 20% trichloroacetic acid to precipitate proteins. After incubating the samples for 30 min on ice, the mixture was next centrifuged at 20,000× *g* for 30 min. The pellet was washed with acetone to remove acid and centrifuged again at 20,000× *g* but for 15 min. After removing the supernatant, the pellet was resuspended in 2X loading buffer containing 63 mM Tris (pH 6.8), 10% glycerol, 2% SDS, 0.0013% bromophenol, and 5% β-mercaptoethanol. The dissolved proteins were denatured at 100 °C for 5 min.

### 4.4. Enzyme-Linked Immunosorbent Assay (ELISA)

SweAPP/N2a cells were placed on culture plates. After overnight incubation, the cells were serum-starved and treated with verbenalin for 24 h. The Aβ 40 and Aβ 42 levels in the cell supernatant were measured using a commercial ELISA kit (IBL, Minneapolis, MN, USA), according to the manufacturer’s protocol. Absorbance was measured at 450 nm using a multimode reader, SpectraMAX i3x (Molecular Devices, San Jose, CA, USA).

### 4.5. Animal Care

All animal care procedures were approved by the Institutional Animal Care and Use Committee (IACUC) guidelines of Woosuk University (IACUC WS2020-06). Animals were maintained in a light-controlled environment (reversed 12 h light/dark cycle, with lights turned on at 7:00 a.m.). Food and water were provided ad libitum. In the present experimental study, male B6;C3-Tg(APPswe, PSEN1dE9)85Dbo/Mmjax (Jackson Laboratory, Bar Harbor, ME, USA) and C57BL/6J mice were randomly allocated into five groups (Group 1: normal C57BL/6 control mice + 0.9% saline; Group 2: B6Cg-Tg AD mice + 0.9% saline; Group 3: B6Cg-Tg AD mice + verbenalin (100 mg/kg); Group 4: B6Cg-Tg AD mice + verbenalin (200 mg/kg); Group 5: B6Cg-Tg AD mice + donepezil (5 mg/kg)). Verbenalin and donepezil were administered once daily for 35 d via oral gavage using a stainless-steel needle. The mice were then euthanized on day 36.

### 4.6. Immunohistochemistry (IHC)

The sectioned brain tissues were attached to glass slides and washed with 0.1 M PBS for 15 min. The sections were then incubated for 30 min with 0.1 M PBS + 1% BSA + 0.2% Triton X-100 + 0.3% H_2_O_2_ + 1.5% normal goat serum. After washing a second time with 0.1 M PBS, the sections were incubated with Aβ (1:400, ab2539, Abcam), tau (1:200, ab151559, Abcam) and BDNF (1:150, ab108319, Abcam) at 4 °C. After overnight incubation, they were washed twice again with 0.1 M PBS for 10 min and then the incubated with secondary antibodies for 1 h. IHC staining was performed using the VECTASTAIN^®^ Elite ^®^ ABC-HRP Kit (PK-6100, Vector Laboratories, Inc., Burlingame, CA, USA) for 30 min. After washing, visualization was performed with an 3-3′diaminobenzidine (D4293, Sigma-Aldrich) for 10 min. The sections were then imaged under a microscope. Slides were evaluated for relative staining intensity and distribution within the hippocampus (scale 0 to +++) and the results were recorded as follows: - negative, + weak positive, ++ moderate positive, and +++ strong positive.

### 4.7. Statistical Analysis

Statistical analysis was performed using SigmaPlot software 14.5 (Systat Software Inc., San Jose, CA, USA). All experimental data are expressed as the mean ± SEM. Significant differences were assessed using Student’s *t*-test or one-way analysis of variance (ANOVA). Asterisks indicate statistically significant differences between groups: * *p* < 0.05, ** *p* < 0.01, and *** *p* < 0.001.

## 5. Conclusions

Among the primary focuses of treating AD should be the accumulation of Aβ and tau phosphorylation in the neurodegenerative processes associated with AD. Our findings demonstrate that verbenalin treatment decreased the generation of Aβ peptides in both in vitro and in vivo AD models. Additionally, verbenalin restored the neurotrophic factor BDNF, which was reduced in the hippocampal brain region of AD animal models. Verbenalin may thus be utilized as a potential supplement or at least as an adjuvant medication to prevent and manage disorders such as AD.

## Figures and Tables

**Figure 1 molecules-27-08678-f001:**
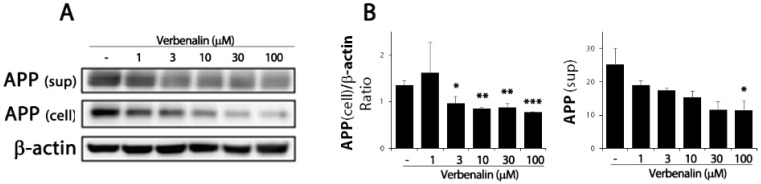
Effects of verbenalin on APP expression in SweAPP/N2a cells. (**A**) SweAPP/N2a cells were treated with the indicated verbenalin concentration for 24 h. APP protein levels in cell lysate and supernatant were analyzed by Western blotting. β-actin was used as loading control. (**B**) Data were quantified by densitometric analysis. Results are represented as the mean ± SEM. * *p* < 0.05, ** *p* < 0.01, and *** *p* < 0.001 by one-way analysis of variance (ANOVA).

**Figure 2 molecules-27-08678-f002:**
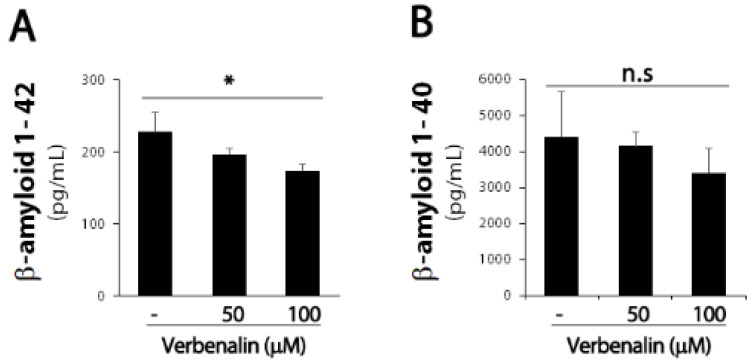
Effects of verbenalin on Aβ peptides generation in SweAPP/N2a cells. (**A**) Swe APP/N2a cells were treated with the indicated verbenalin concentration for 24 h. The levels of Aβ 42 (**A**) and Aβ 40 (**B**) were determined using ELISA. Data were quantified by densitometric analysis. Results are represented as the mean ± SEM. * *p* < 0.05 by Student’s *t*-test.

**Figure 3 molecules-27-08678-f003:**
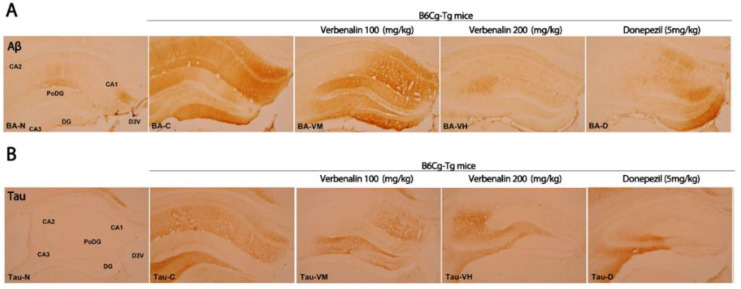
Effects of verbenalin on Aβ- and tau-positive fibers in hippocampus. Aβ (**A**) and tau (**B**) expression is examined using IHC. N: normal group (C57BL/6J mice), C (B6Cg-Tg mice): control group, D: donepezil group, VM: verbenalin medium concentration group (100 mg/kg), VH: verbenalin high concentration group (200 mg/kg), CA1: cornu ammonis area 1, CA2: cornu ammonis area 2, CA3: cornu ammonis area 3, PoDG: polymorphic layer of the dentate gyrus, and DG: dentate gyrus.

**Figure 4 molecules-27-08678-f004:**
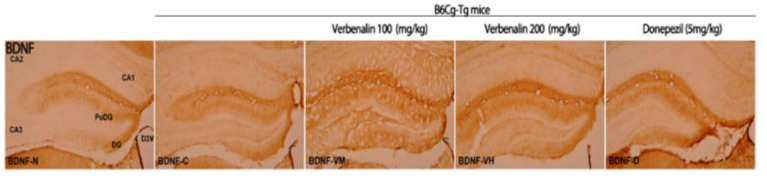
Effects of verbenalin on BDNF-positive fibers in hippocampus. BDNF expression is examined using IHC. N: normal group (C57BL/6J mice), C (B6Cg-Tg mice): control group, D: donepezil group, VM: verbenalin medium concentration group (100 mg/kg), VH: verbenalin high concentration group (200 mg/kg), CA1: cornu ammonis area 1, CA2: cornu ammonis area 2, CA3: cornu ammonis area 3, PoDG: polymorphic layer of the dentate gyrus, and DG: dentate gyrus.

**Table 1 molecules-27-08678-t001:** The change in Aβ immunoreactive neurons of hippocampus. Staining density: - negative, + weak positive, ++ moderate positive, and +++ strong positive. CA1: cornu ammonis area 1, CA2: cornu ammonis area 2, CA3: cornu ammonis area 3, PoDG: polymorphic layer of the dentate gyrus, and DG: dentate gyrus.

	C57BL/6	B6Cg-Tg			
	-	-	Verbenalin 100 mg/kg	Verbenalin 200 mg/kg	Donepezil 5 mg/kg
CA 1	+	+++	+++	-	+
CA 2	+	++	-	-	-
CA 3	-	+	+	+	-
PoDG	-	+++	+++	-	+
DG	-	+++	+++	-	+++
Total	2	12	4	1	5

**Table 2 molecules-27-08678-t002:** The change in tau immunoreactive neurons of hippocampus. Staining density: - negative, + weak positive, ++ moderate positive, and +++ strong positive. CA1: cornu ammonis area 1, CA2: cornu ammonis area 2, CA3: cornu ammonis area 3, PoDG: polymorphic layer of the dentate gyrus, DG: dentate gyrus.

	C57BL/6	B6Cg-Tg			
	-	-	Verbenalin 100 mg/kg	Verbenalin 200 mg/kg	Donepezil 5 mg/kg
CA 1	-	++	+	-	-
CA 2	-	++	-	-	-
CA 3	-	+++	++	++	+++
PoDG	-	++	-	-	+
DG	-	+	+	-	-
Total	0	10	4	2	4

**Table 3 molecules-27-08678-t003:** The change of BDNF immunoreactive neurons of hippocampus. Staining density; -: Negative, +: Weak positive, ++: Moderate positive, +++: Strong positive. CA1: Cornu Ammonis Area 1, CA2: Cornu Ammonis Area 2, CA3: Cornu Ammonis Area 3, PoDG: Polymorphic layer of the dentate gyrus, DG: Dentate gyrus.

	C57BL/6	B6Cg-Tg			
	-	-	Verbenalin 100 mg/kg	Verbenalin 200 mg/kg	Donepezil 5 mg/kg
CA 1	-	-	+	+	-
CA 2	-	-	-	-	-
CA 3	-	-	+	+	-
PoDG	++	+	++	++	++
DG	++	+	++	++	++
Total	4	2	6	6	4

## Data Availability

The data presented in this study are available in this article.

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
