# Peer review of "Verbenalin Reduces Amyloid-Beta Peptide Generation in Cellular and Animal Models of Alzheimer’s Disease"

_molecules, 2022, doi:10.3390/molecules27248678_

Round 1

Reviewer 1 Report

The manuscript showed verbenalin may decrease amyloid-beta (Aβ)  formation both in vitro and in vivo. Overall, the manuscript is technically sound and the research ideas appears justified.Listed are some comments regarding the submitted manuscript:

1.      Line 47-48: Were there any research previously the effect of verbenalin on Alzheimer’s disease (AD)?

This paper should be accepted for publication after minor revision.

Author Response

We would like to express our sincere appreciation to the reviewers’ comments to improve our manuscript. We have tried our best to resolve each of the comments and hope the revision is satisfactory. We provide our point-to-point responses here and the changes made are shown in red-colored font in the revised manuscript.

  1. “Line 47-48: Were there any research previously the effect of verbenalin on Alzheimer’s disease (AD)?” Thank you for your kind comment. Other than the microarray and cytotoxicity evaluation using hAEC cells as mentioned in the manuscript (PMID: 32224504), no studies using verbenalin have been conducted on Alzheimer's disease. Further, the findings of a study showing that the aqueous extracts of Verbena officinalis reduce Aβ-induced neurotoxicity in primary cortical neurons have been added in line 56-58 (PMID: 16406021).

Reviewer 2 Report

Dear Authors:

The manuscript "Verbenalin reduces amyloid-beta peptide generation in cellular and animal models of Alzheimer’s disease" by Lim et al. has demonstrated that  verbenalin may decrease Aβ formation both in vitro and in vivo. Verbenalin may also help to improve the pathological hallmarks of AD. I have just a few suggestions.

1. The manuscript needs linguistic improvement.

2. Some background information or references are missing. In introduction, please add more background information about the pathogenesis of AD, such as cerebral vascular dysfunction, which plays a more and more important role in AD. Some reviews has demonstrated it. (please cite: 1. From 1901 to 2022, how far are we from truly understanding the pathogenesis of age-related dementia? Geroscience. 2022 Jun;44(3):1879-1883. doi: 10.1007/s11357-022-00591-7.

2. Novel Mechanistic Insights and Potential Therapeutic Impact of TRPC6 in Neurovascular Coupling and Ischemic Stroke. Int J Mol Sci. 2021 Feb 19;22(4):2074. doi: 10.3390/ijms22042074.)

Best,

Author Response

  1. “The manuscript needs linguistic improvement.” Thank you for your kind comment. The manuscript has been checked by an editing service and the English proofreading certificate is attached herewith.
  2. “Some background information or references are missing. In introduction, please add more background information about the pathogenesis of AD, such as cerebral vascular dysfunction, which plays a more and more important role in AD. Some reviews has demonstrated it. (please cite: 1. From 1901 to 2022, how far are we from truly understanding the pathogenesis of age-related dementia? Geroscience. 2022 Jun;44(3):1879-1883. doi: 10.1007/s11357-022-00591-7, 2. Novel Mechanistic Insights and Potential Therapeutic Impact of TRPC6 in Neurovascular Coupling and Ischemic Stroke. Int J Mol Sci. 2021 Feb 19;22(4):2074. doi: 10.3390/ijms22042074.)” Thank you for your careful comment. We have rewritten the introduction section and added recommended the references regarding cerebral vascular dysfunction. (Line 45-53).

Reviewer 3 Report

Verbenaline as well as thousands of molecules extracted from nature are certainly an inexhaustible source of compound leader. But the lack of selectivity in terms of pharmacodynamics cannot be counted as a therapeutic means as it lacks the safety protocols imposed by the international or national pharmacopoeia in which it could be used. nevertheless. therefore, if this work is well conducted both in terms of histochemical and molecular expression, it must be supported by specialists in the medicinal chemistry sector so as to engineer, starting from this lead, a selective molecule for the inhibition of the accumulation of Aβ and tau phosphorylation in the neurodegenerative processes associated with AD. Therefore, due to the skills conferred on me in this area, I deem it necessary to remember that this crude molecule can only be used as a possible dietary supplement.

Author Response

  1. “Verbenaline as well as thousands of molecules extracted from nature are certainly an inexhaustible source of compound leader. But the lack of selectivity in terms of pharmacodynamics cannot be counted as a therapeutic means as it lacks the safety protocols imposed by the international or national pharmacopoeia in which it could be used. nevertheless. therefore, if this work is well conducted both in terms of histochemical and molecular expression, it must be supported by specialists in the medicinal chemistry sector so as to engineer, starting from this lead, a selective molecule for the inhibition of the accumulation of Aβ and tau phosphorylation in the neurodegenerative processes associated with AD. Therefore, due to the skills conferred on me in this area, I deem it necessary to remember that this crude molecule can only be used as a possible dietary supplement.” Thank you for your thoughtful comment. We agree that verbenalin can be used as an adjuvant therapy to slow the neurodegenerative processes associated with AD. In fact, the verbenalin used in this study was obtained from a commercial source rather than as a crude molecule. We have rewritten our conclusions to reflect this point (Line 198-204).

Round 2

Reviewer 3 Report

MEDICINAL CHEMISTRY is not clear to the authors. the same introductory lines that I write to you again are:

 Medicinal Chemistry

A section of Molecules (ISSN 1420-3049).

Section Information

The Medicinal Chemistry Section of the journal Molecules publishes original research and review articles that increase our understanding of how the chemical structure of bioactive molecules determines their pharmacodynamic, pharmacokinetic, and physicochemical properties and, hence, their therapeutic potential. In particular, this Section mainly invites contributions that report on:

-The design, synthesis, chemical characterization1, and biological evaluation of novel compounds against biological targets of therapeutic, diagnostic, or theranostic interest, and/or the assessment of their physicochemical and pharmacokinetic properties;

-The isolation from herbal plants and separate biological characterization of individual bioactive compounds and/or their semisynthetic derivatives;

-Hit-to-lead projects that lead to a significant optimization of potency, or physicochemical and/or pharmacokinetic properties;

-Computational2, molecular biology, and/or structural biology studies that shed light on the structure and function of biological targets of therapeutic, diagnostic, or theranostic interest and on their interaction with known or novel compounds, thereby enabling the rational design of optimized ligands;

-Computational2 and biological studies of known drugs or their derivatives that address their interaction with novel biological targets for repositioning purposes;

-Novel synthetic, computational2, or biological methodologies of relevance in medicinal chemistry;

-Development and application of materials for medicinal chemistry purposes, such as drug delivery and drug targeting.

Manuscripts that report on the biological evaluation of mixtures of compounds, such as plant extracts, or purely pharmacological studies that look deeper into the standard action of well-established drugs will normally not be considered for publication in this Section.

 but none of these come close to the proposed work. If anything it is only a pharmacological work, therefore not suitable for this section.